# The aryl hydrocarbon receptor controls cyclin O to promote epithelial multiciliogenesis

Matteo Villa[1,*], Stefania Crotta[2,*], Kevin S. Dingwell[3], Elizabeth M.A Hirst[4], Manolis Gialitakis[1], Helena Ahlfors[5], James C. Smith[3], Brigitta Stockinger[1,*] & Andreas Wack[2,*]

Epithelia function as barriers against environmental insults and express the transcription factor aryl hydrocarbon receptor (AhR). However, AhR function in these tissues is unknown. Here we show that AhR regulates multiciliogenesis in both murine airway epithelia and in *Xenopus laevis* epidermis. In air-exposed airway epithelia, induction of factors required for multiciliogenesis, including cyclin O (Ccno) and Multicilin (Mcidas), is AhR dependent, and air exposure induces AhR binding to the *Ccno* promoter. Submersion and hypoxic conditions impede AhR-dependent *Ccno* induction. This is mediated by the persistence of Notch signalling, as Notch blockade renders multiciliogenesis and *Ccno* induction by AhR independent from air exposure. In contrast to *Ccno* induction, air exposure does not induce the canonical AhR target cytochrome P450 1a1 (*Cyp1a1*). Inversely, exposure to AhR ligands induces *Cyp1a1* but not *Ccno* and impeded ciliogenesis. These data indicate that AhR involvement in detoxification of environmental pollutants may impede its physiological role, resulting in respiratory pathology.

[1] Laboratory of AhRimmunity, The Francis Crick Institute, Mill Hill Laboratory, The Ridgeway, London NW7 1AA, UK. [2] Laboratory of Immunoregulation, The Francis Crick Institute, Mill Hill Laboratory, The Ridgeway, London NW7 1AA, UK. [3] Laboratory of Developmental Biology, The Francis Crick Institute, Mill Hill Laboratory, The Ridgeway, London NW7 1AA, UK. [4] EM Technology Platform, The Francis Crick Institute, Mill Hill Laboratory, The Ridgeway, London NW7 1AA, UK. [5] Lymphocyte Signalling and Development, The Babraham Institute, Cambridge CB22 3AT, UK. * These authors contributed equally to this work. Correspondence and requests for materials should be addressed to A.W. (email: andreas.wack@crick.ac.uk).

Epithelial tissues represent important barriers shielding the organism from invading pathogens and harmful substances[1]. In the respiratory tract, the combined action of epithelial multiciliated cells and secretory cells generate the mucociliary escalator required for the clearance of microbes and particles from the airways. Defects in any of the components of mucociliary clearance are associated with lung damage, frequent infections and chronic inflammation[2]. The development and frequency of the different epithelial cell types responsible for mucociliary flow are therefore tightly controlled[1,3,4].

The development of multiciliated cells from basal cells in the airway epithelium involves multiplication of centrioles in a cell cycle-independent manner and the orientation of the centrioles towards the apical cell surface to form basal bodies, from which the ciliar axoneme develops[3]. Factors required for multiciliogenesis, including Multicilin (Mcidas)[5], GemC1 (refs 6,7), cyclin O (Ccno)[8,9], and the transcription factors p73 (refs 10,11) and Foxj1 (refs 12,13) are upregulated and maintained in a transcriptional network in cells committed to the multiciliated lineage[3]. How multiciliation is influenced by environmental cues is however incompletely understood.

The aryl hydrocarbon receptor (AhR) is a transcription factor expressed widely in epithelia, particularly those of barrier organs such as the skin, gut and lung[14,15]. AhR is best known for mediating the effects of toxins such as 2,3,7,8-tetrachlorodibenzo-$p$-dioxin, and many air pollutants such as vehicle exhaust and cigarette smoke contain potent AhR agonists[16]. Binding of these toxicological agonists to AhR triggers its nuclear translocation and targeting, and induction of genes encoding drug metabolizing enzymes of the cytochrome P450 family, thus triggering a detoxification programme. While physiological roles of AhR in the mammalian immune system have recently started to emerge (reviewed in ref. 14), less is known about AhR function in epithelia[17].

Here we describe a novel and evolutionary conserved role for AhR in the promotion of multiciliogenesis in murine airway epithelia and in *Xenopus laevis* epidermis. Among the genes involved in orchestrating multiciliogenesis, we identified cyclin O[8,9] as direct AhR target. While air exposure caused the AhR-dependent induction of *Ccno*, it did not induce the canonical AhR target gene cytochrome P450 1a1 (*Cyp1a1*). In contrast, exposure to potent AhR ligands induced *Cyp1a1* but not *Ccno* in epithelia, indicating that AhR is able to target different transcriptional programmes depending on the environmental trigger. Importantly, non-degradable AhR ligands impeded ciliogenesis. Our data suggest that interference between ligand-induced detoxifying AhR functions and its role in multiciliogenesis may be of pathophysiological relevance in the context of environmental pollution that is known to adversely affect respiratory functions.

## Results

**Multiciliogenesis is impaired in the absence of AhR.** The development of airway epithelia can be studied in cultured mouse tracheal epithelial cells (mTECs)[18], which recapitulate the differentiation of distinct epithelial cell types from pluripotent progenitors. In particular, exposure of cultured mTECs to air causes the formation of multiciliated cells, which are crucial for airway mucociliary flow[2] and can be identified by staining for acetylated α-tubulin, a major component of the axoneme. We discovered a significant reduction in the number of multiciliated cells and in the overall acetylated α-tubulin signal in mTEC cultures derived from AhR-deficient mice[19] after 9 days of exposure to air (Fig. 1a–c, with Fig. 1c showing signal quantification from micrographs of whole mTEC transwells as found in Supplementary Fig. 1a). Scanning electron microscopy

showed that wild-type multiciliated cells had a regular pattern of cilia covering most of the cell surface, while cilia were sparsely distributed on the few $Ahr^{-/-}$ epithelial cells that did form cilia, with most of the cell surface left undecorated (Fig. 1d). Since each cilium has at its base a basal body, which is crucial for ciliar organization, the multiplication of centrioles is an important step in multiciliogenesis[20]. γ-Tubulin staining of basal bodies (Fig. 1e) showed basal body multiplication and localization at the ciliar base on the apical side of the epithelia in wild-type cells. This process was strongly impaired in AhR-deficient epithelial cells, where fewer basal bodies were present that docked at the apical side (Fig. 1e, z axis).

Overall, AhR deficiency causes both a reduction in the number of multiciliated cells and a defect in their morphogenesis, with fewer basal bodies developing and docking to the apical surface to extend cilia. To better distinguish these two components of the phenotype, we stained epithelia for the transcription factor Foxj1 that accumulates just before ciliogenesis in cells committed to the multiciliated fate, and for acetylated α-tubulin to visualize ciliar structure. We found both a reduced frequency of Foxj1$^+$ cells (Fig. 2a,c) and among Foxj1$^+$ cells an increased number of cells with disorganized cilia (uneven cilia distribution on the apical surface and acetylated α-tubulin staining extending to the basolateral regions of the cell; Fig. 2a,b,d) in $Ahr^{-/-}$ compared with wt mTECs. This compound phenotype recapitulates that observed in epithelial cells of patients with *CCNO* and *MCIDAS* mutations and of mice with targeted *Ccno* or *Foxj1* disruption[8,9,13,21].

**AhR requirement for multiciliogenesis *in vivo*.** We next asked whether AhR plays a role in ciliogenesis *in vivo*. Many multiciliated cells in E18.5 $Ahr^{-/-}$ mutant tracheae had a lower density or patchy distribution of cilia as compared with analogous cells in wild-type embryos (Fig. 3a,b). However, this was not true of tracheae from adult AhR-deficient mice (Supplementary Fig. 1b). This observation is similar to the transitional phenotype in Myb-deficient mice[22] and in the Ccno-deficient mouse model of human ciliopathy[8], and suggests that multiple mechanisms are employed to safeguard multiciliation in the developing lung.

AhR is highly conserved in evolution[23]. We therefore tested its function in an evolutionary distant model system, *Xenopus laevis*. Like the airway, the epidermis of amphibian embryos develops as a mixture of mucus-secreting goblet cells, ionocytes and multiciliated cells[24,25]. In *Xenopus*, multiciliated cells are derived from epithelial sublayer precursors that undergo radial intercalation into the outer epithelium, thereby forming a punctate pattern of ciliated cells (reviewed in ref. 25). Targeting of the two paralogues of AhR present in this species[26] with antisense morpholino oligonucleotides caused defects in ciliogenesis and trafficking of basal bodies to the apical surface of multiciliated cells (Fig. 3c,d), reminiscent of the phenotype observed in $Ahr^{-/-}$ mTEC cultures (Fig. 1a–e). Targeting of the single AhR paralogues had no effect (Supplementary Fig. 2), indicating functional redundancy of Ahr1α and β in *Xenopus*. In contrast to mTECs, we did not find a reduction in cells committed to the multiciliated lineage (Fig. 3e), as identified by multiplied basal bodies and the presence of acetylated α-tubulin staining. We found however acetylated α-tubulin mainly within the cell body, not in ciliar patterns above the apical surface of the cells (Fig. 3c,d), indicative of a defect in ciliogenesis. From these results, we conclude that AhR regulates epithelial multiciliogenesis in organisms as distant in evolution as *Xenopus* and mice.

**AhR directly targets *Ccno* to induce multiciliogenesis.** To gain further insights into this process, we analysed mTEC cultures at

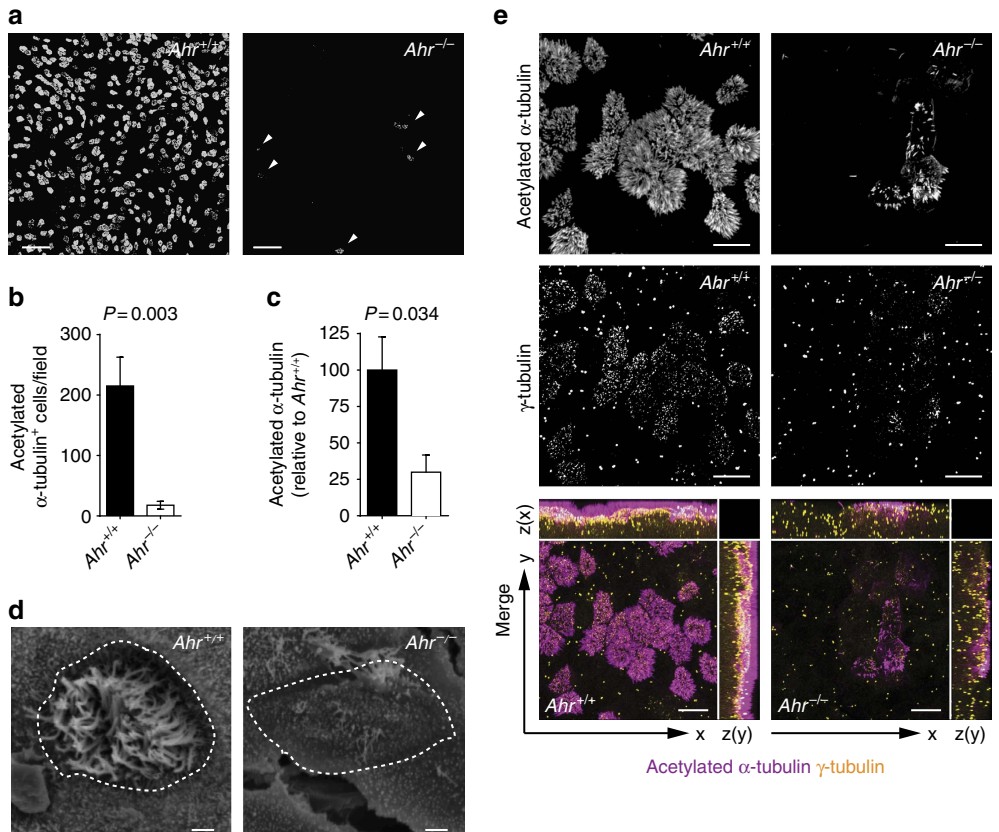

**Figure 1 | Multiciliated cell generation is markedly reduced in AhR-deficient mTEC cultures.** (**a**) Immunofluorescent staining of acetylated α-tubulin in AhR-sufficient and AhR-deficient mTEC cultures following 9 days of ALI. Arrowheads indicate the few multiciliated cells (acetylated α-tubulin$^+$) developing from $Ahr^{-/-}$ cultures. Scale bars, 50 μm. (**b**) Number of acetylated α-tubulin$^+$ cells per field in wt and $Ahr^{-/-}$ mTEC cultures after 9 days of ALI. Mean ± s.e.m.; Student's $t$-test (unpaired, two-tailed). $n = 5$ random fields per group. (**c**) Quantification of acetylated α-tubulin fluorescence intensity from entire wells of AhR-sufficient and AhR-deficient mTEC cultures at day 9 of ALI. Fluorescence intensity of $Ahr^{+/+}$ condition was set as 100%. $n = 3$ wells per group. Mean ± s.e.m.; Student's $t$-test (unpaired, two-tailed). Data in **a–c** are from five independent experiments. (**d**) Scanning electron microscopy (SEM) images of AhR-sufficient and AhR-deficient mTEC cultures at 9 days of ALI. Dashed lines indicate the multiciliated cell perimeter. Scale bars, 1 μm. Data representative of two independent experiments. (**e**) Immunofluorescent staining of acetylated α-tubulin (cilia, violet) and γ-tubulin (centrioles/basal bodies, yellow) in AhR-sufficient and AhR-deficient mTEC cultures at 9 days of ALI. Scale bars, 10 μm. Data representative of three independent experiments.

different times after exposure to air (air–liquid interphase, ALI). AhR was present in wild-type mTECs at the onset of ALI and increased thereafter (Fig. 4a,b; Supplementary Fig. 3). A selection of genes involved in multiciliogenesis that were upregulated in wild-type mTECs failed to be activated upon exposure to air in $Ahr^{-/-}$ mTECs, including *Ccno*, *Mcidas*, *Myb* and *Ccdc67* (encoding for the deuterosomal protein Deup1; reviewed in ref. 27; Fig. 4c).

We next asked whether the above genes could be transcriptional targets of AhR. Chromatin immunoprecipitation followed by PCR, using primers for the *Ccno* promoter (Supplementary Fig. 4a,c), identified AhR binding to *Ccno* in the developing airway epithelium upon air exposure (Fig. 4d), suggesting that AhR regulates multiciliogenesis by direct transcriptional activation of one or several key target genes involved in this process. To demonstrate directly that AhR drives *Ccno* expression, luciferase reporter assays were performed in AhR-sufficient or -deficient mTECs. A 1.5 kb DNA fragment upstream of the *Ccno* gene containing putative AhR response elements (*Ccno* promoter) drove luciferase expression only in wild-type cells. In contrast, either AhR deficiency or a truncated promoter (Delta) lacking AhR response elements prevented luciferase expression (Fig. 4e). Similarly, AhR overexpression in NIH 3T3 cells drives *Ccno*

promoter-dependent luciferase expression only when the putative AhR response elements are present (Fig. 4f). On activation and nuclear translocation, AhR forms heterodimers with the nuclear protein ARNT to function as transcriptional activator. Chromatin immunoprecipitation–PCR performed with an anti-ARNT monoclonal antibody established increased *Ccno* targeting by the AhR partner ARNT upon air exposure, suggesting that AhR and ARNT bind the *Ccno* promoter as a dimer, as described for *Cyp1* genes (Supplementary Fig. 4e). In contrast, *Mcidas* appears not to be a direct AhR target (Supplementary Fig. 4d,f).

In mammals as well as in *Xenopus*, the *Ccno* gene is adjacent to the genes *Mcidas* and *Cdc20b*, which incorporates the miR-449 family in intron 2 (Supplementary Fig. 4a,b). These microRNAs are highly expressed in multiciliated epithelia and are important in targeting the multiciliogenesis inhibitors Notch and Cp110 (refs 28,29). We found that *Cdc20b* and *mir449c* were upregulated upon air exposure in an AhR-dependent manner (Fig. 4g). We conclude that AhR promotes ciliogenesis in airway epithelia by inducing *Ccno* and by potentially rendering the locus permissive for expression of nearby genes also involved in ciliogenesis. The induction of a wide range of genes, including the upstream transcription factor Mcidas (Fig. 4c), suggests that AhR may be a central regulator of multiciliogenesis.

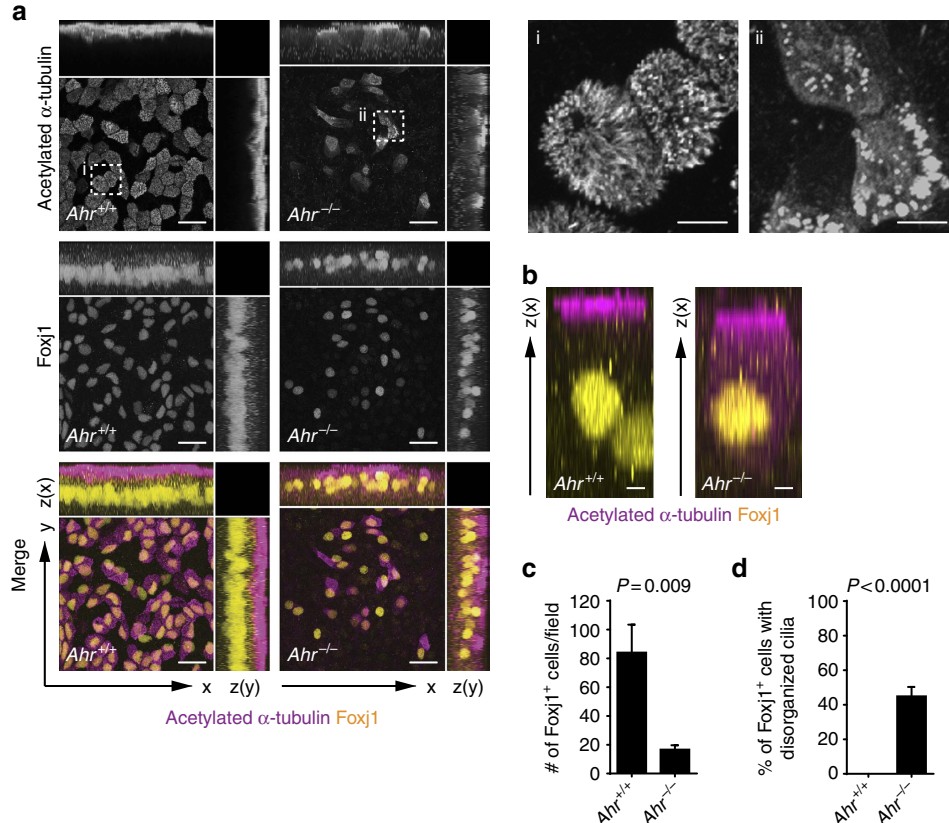

**Figure 2 | AhR deficiency affects both multiciliated cell commitment and cilia organization. (a)** Immunofluorescent staining of acetylated α-tubulin (cilia, violet) and Foxj1 (yellow) in wt and $Ahr^{-/-}$ mTEC cultures at 9 days of ALI. Scale bars, 20 μm. Dashed boxes indicate insets (**i,ii**). Scale bars, 5 μm (for insets **i** and **ii**). Data representative of two independent experiments. (**b**) Immunofluorescent staining of acetylated α-tubulin (cilia, violet) and Foxj1 (yellow) in $Ahr^{+/+}$ and $Ahr^{-/-}$ mTEC cultures following 9 days of ALI. Caption of the z projection of a single multiciliated cell showing distribution of the acetylated α-tubulin staining in the cytoplasm. Scale bars, 2 μm. Data representative of two independent experiments. (**c**) Number of Foxj1$^+$ cells per field in AhR-sufficient and AhR-deficient mTEC cultures at day 9 of ALI. Mean ± s.e.m.; Student's $t$-test (unpaired, two-tailed). $n = 5$ random fields per group, data from two independent experiments. (**d**) Fraction of Foxj1$^+$ cells bearing a disorganized pattern of cilia (uneven distribution on the apical surface; intracytosolic staining of acetylated α-tubulin) in $Ahr^{+/+}$ and $Ahr^{-/-}$ mTEC cultures after 9 days of ALI. Mean ± s.e.m.; Student's $t$-test (unpaired, two-tailed). $n = 5$ random fields per group, data from two independent experiments.

**AhR converts environmental cues into specific transcription**. To understand in more detail the environmental conditions leading to AhR-dependent ciliogenesis, we studied the differentiation of ciliated cells in submersion or in hypoxic conditions. Culturing mTECs under these conditions led to reduced multiciliogenesis (Fig. 5a), as reported[30–32], and blocked *Ccno* induction (Fig. 5b). Consistently, AhR targeted *Ccno* only in normoxia, not submersion or hypoxia (Fig. 5c), indicating that normoxic air exposure is required for AhR to bind to the *Ccno* promoter.

Previous studies of mouse airway and *Xenopus* epidermis models showed that persistent Notch signalling blocks the development of ciliated cells and favours the development of secretory cells[33–35]. To explore the relative roles of Notch and AhR in ciliogenesis, we treated wild-type or $Ahr^{-/-}$ mTEC cultures with the γ-secretase inhibitor DAPT (N-[(3,5-difluorophenyl) acetyl]-L-alanyl-2-phenyl]glycine-1,1-dimethylethyl ester) to block Notch signalling. In wild-type mTECs, this led to an increased number of multiciliated cells as measured by the expression of acetylated α-tubulin (Fig. 5d), as previously described[33,34], and to the upregulation of ciliogenesis-related genes such as *Ccno* (Fig. 5e). In contrast, in $Ahr^{-/-}$ mTECs, the development of multiciliated cells was not induced, and *Ccno* was not upregulated by Notch inhibition (Fig. 5d,e),

indicating that AhR acts downstream of Notch. It has been shown previously that blocking Notch signalling releases the inhibition of ciliogenesis observed in submerged culture[30]. When we treated submerged epithelia with DAPT to block Notch signalling, *Ccno* targeting by AhR as well as ciliogenesis were restored (Fig. 5f; Supplementary Fig. 5a,b). These results demonstrate that persistence of Notch signalling inhibits AhR-mediated *Ccno* induction and suggest that AhR targeting of the *Ccno* promoter requires Notch blockade.

To understand the potential relationship between agonist-driven AhR activation and *Ccno* targeting, we compared the induction of detoxifying AhR target genes such as *Cyp1a1* versus *Ccno* by different stimuli. *Ccno* was induced by exposure to air, but not by the known AhR agonist 6-formylindolo [3,2-b]carbazole (FICZ)[36,37] (Fig. 6a). In contrast, FICZ, but not exposure to air, led to induction of *Cyp1a1* (Fig. 6b). This suggests that the activation of different AhR target genes depends on different environmental cues, with potent agonists inducing detoxifying proteins such as CYP1 family members, and exposure to air activating genes important in ciliogenesis.

Given the fact that AhR activation via a potent ligand did not induce *Ccno*, we suspected that exposure of developing

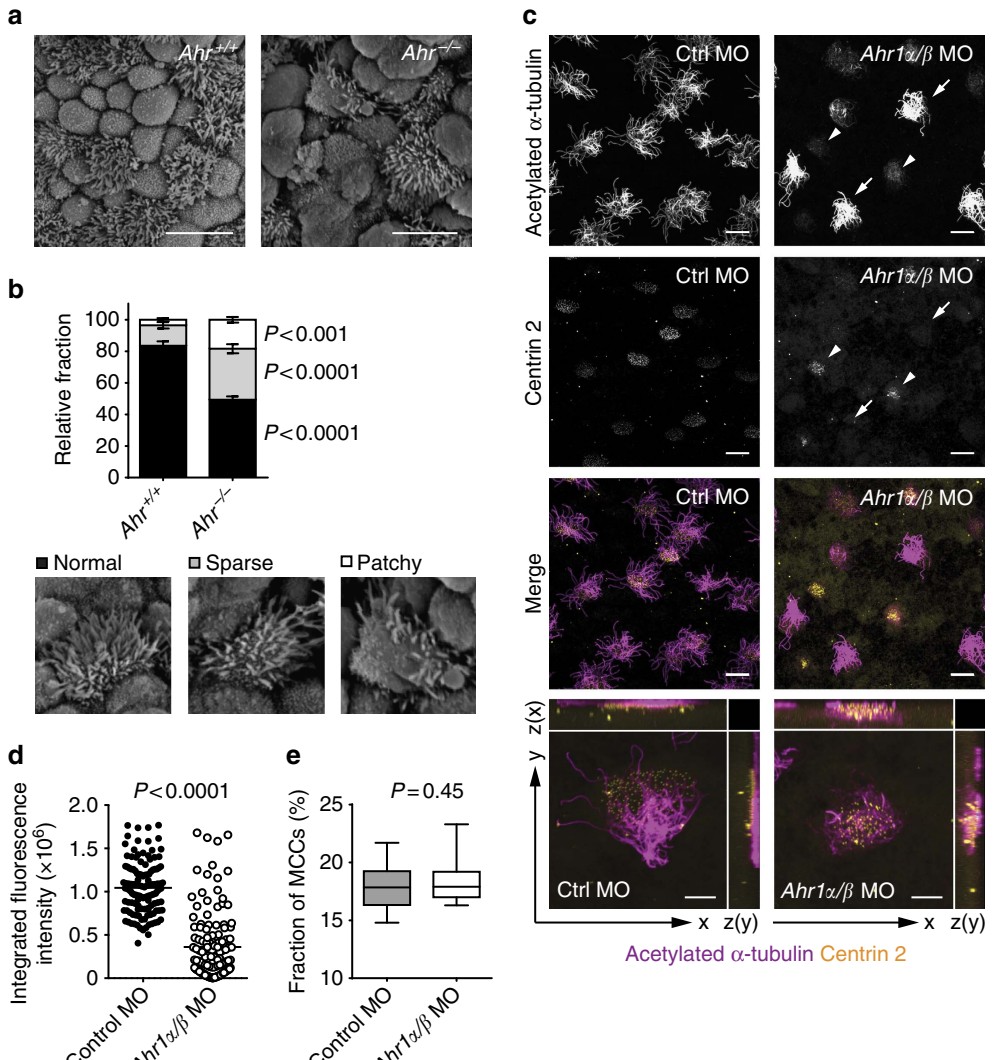

**Figure 3 | Deletion of *Ahr* leads to an aberrant cilia pattern in both mouse and *Xenopus laevis* embryos.** (**a**) SEM images of tracheae from $Ahr^{+/+}$ and $Ahr^{-/-}$ E18.5 embryos. Scale bars, 5 µm. (**b**) Relative fraction of multiciliated cells in E18.5 embryonic tracheae as in **a**, classified by their ciliar pattern (normal, sparse and patchy). No significant differences were observed in the number of multiciliated cells per field between the two genotypes. Mean ± s.e.m.; two-way analysis of variance (variables: genotype and cilia pattern) with Sidak's multiple comparison test. Data from two independent litters ($n = 5$–9). (**c**) Multiciliated cells containing $Ahr1\alpha/\beta$ morpholinos ($Ahr1\alpha/\beta$ MO—arrowheads) in $Ahr1\alpha/\beta$ MO-injected embryos develop very few cilia, and basal bodies fail to migrate and dock at the apical surface, compared either with cells that are uninjected (arrows) or with cells in embryos injected with control MO. Scale bars, 15 µm. Bottom panels show the accumulation of acetylated α-tubulin above the apical surface of the cell in control morphants, while in $Ahr1\alpha/\beta$ morphants, microtubules amass below the apical surface. Scale bars, 5 µm. (**d**) Quantification of integrated fluorescence intensity of acetylated α-tubulin above the apical surface of multiciliated cells, from the embryonic skin of *X. laevis* injected with $Ahr1\alpha/\beta$ or control MO. Mean; Student's *t*-test (unpaired, two-tailed). Data representative of three independent experiments. (**e**) Fraction of multiciliated cells, from the embryonic skin of *X. laevis* injected with $Ahr1\alpha/\beta$ or control MO. Mean ± maximum and minimum; Student's *t*-test (unpaired, two-tailed). Data representative of three independent experiments. Ctrl, control; MCCs, multiciliated cells.

airway epithelia to AhR activation by environmental agents might interfere with ciliogenesis. This was supported by the finding that exposure of mTEC cultures to the cancerogenic aromatic hydrocarbon 3-methylcholanthrene (3MC), a potent AhR agonist, resulted in reduced induction of *Ccno* (Fig. 6c) and *Mcidas* (Supplementary Fig. 6a), and reduced development of ciliated cells (Fig. 6d; Supplementary Fig. 6b). 3MC exposure reduced both commitments to the multiciliated lineage, as measured by the number of Foxj1$^+$ cells (Fig. 6e,f), and ciliar organization, as assessed by the staining pattern of acetylated α-tubulin (Fig. 6e,g), which recapitulates the phenotype found in AhR deficiency (Fig. 2).

## Discussion

The evolutionary conservation of AhR is an indicator of its physiological functions, which are gradually beginning to emerge, until recently overshadowed by the focus on AhR roles in the detoxification of environmental pollutants.

Here we show that AhR is an evolutionary conserved regulator of multiciliogenesis in epithelia. In both murine airway epithelia and in *X. laevis* epidermis, the absence or reduction of AhR led to a marked reduction in numbers of motile cilia. In mouse tracheal cell cultures, we additionally find a reduction in cells committed to the multiciliated lineage, as shown by the expression of the transcription factor Foxj1, whereas this was not evident when silencing AhR1α/β in *X. laevis* epidermis.

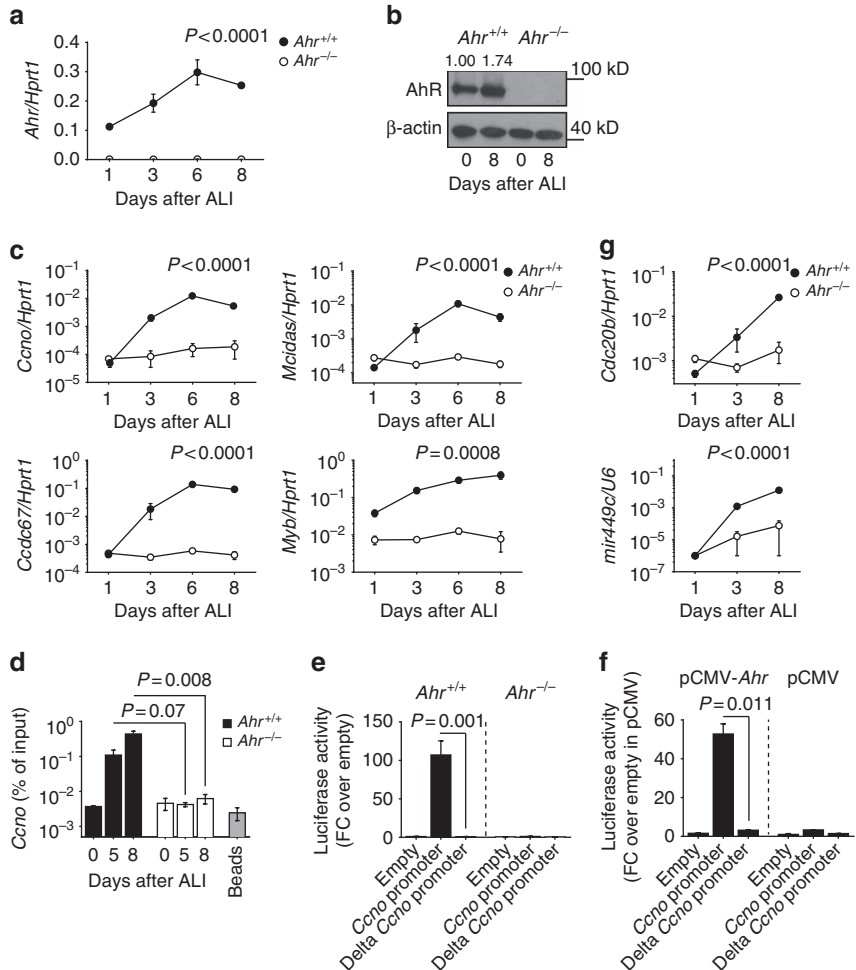

**Figure 4 | AhR controls *Ccno* expression and regulates the transcriptional programme required for ciliogenesis.** (**a**,**c**) mRNA expression levels of the indicated genes in $Ahr^{+/+}$ and $Ahr^{-/-}$ mTEC cultures at indicated days of ALI. Values are normalized to *Hprt1*. Mean ± s.e.m.; two-way analysis of variance (variables: genotype and time). Data representative of at least three independent experiments. (**b**) Immunoblot and densitometric analysis of AhR protein expression as compared with β-actin, in AhR-sufficient and AhR-deficient mTEC cultures at days 0 and 8 of ALI. Data representative of three independent experiments. (**d**) Chromatin immunoprecipitation (ChIP) analysis of AhR interaction with the *Ccno* promoter in wt and $Ahr^{-/-}$ mTEC cultures at indicated days of ALI. Mean ± s.e.m.; Student's *t*-test (unpaired, two-tailed). Data representative of three independent experiments. (**e**) Luciferase reporter assay was performed in AhR-sufficient and AhR-deficient mTEC cultures transfected at 5 days of ALI and analysed 36 h post transfection. Delta *Ccno* promoter lacks predicted AhR response elements. Mean ± s.e.m.; Student's *t*-test. Data representative of two independent experiments. (**f**) Luciferase reporter assay was performed in NIH3T3 cells. Cells were co-transfected with a plasmid encoding AhR (pCMV-*Ahr*) or control vector (pCMV) and a luciferase-encoding reporter gene downstream of the *Ccno* promoter. Cells were analysed 36 h post transfection. Delta *Ccno* promoter lacks predicted AhR response elements. Empty indicates the control reporter vector. Mean ± s.e.m. Student's *t*-test. Data representative of two independent experiments. (**g**) mRNA expression levels of *Cdc20b* (normalized to *Hprt1*) and *mir449c* (normalized to *U6*) in $Ahr^{+/+}$ and $Ahr^{-/-}$ mTEC cultures at indicated days of ALI. Mean ± s.e.m.; two-way analysis of variance (variables: genotype and time). Data are representative of three independent experiments. FC, fold change.

In airway epithelia, the promoter of the multiciliogenesis regulator *Ccno*[5,8,9] was directly induced by AhR during air exposure. In contrast, air exposure did not induce the canonical AhR target gene cytochrome P450 1a1 (*Cyp1a1*). Inversely, exposure to potent AhR ligands such as the tryptophan metabolite FICZ[36] or the carcinogenic aromatic hydrocarbon 3MC[37] induced *Cyp1a1* but not *Ccno* in epithelia, indicating that AhR is able to target different transcriptional programmes depending on the environmental trigger.

Many of the physiological functions ascribed to AhR are important at mucosal surfaces such as the gut, where the maintenance of intraepithelial lymphocytes and interleukin-22-producing cell types depends on AhR signalling[38–41]. In the skin, AhR signalling ameliorates inflammatory responses during induction of psoriasis[42]. Here we show that AhR also

has direct roles in epithelial homeostasis, by promoting multiciliation in epithelial cells.

The AhR-dependent phenotype on ciliogenesis was pronounced *in vitro* and in embryonic stages but not overt in adult airways. This is reminiscent of other gene defects related to multiciliation such as Ccno deficiency[8] which leads to complete absence of cilia in mouse embryos, while adults show a comparably milder phenotype with reduced and disorganized cilia. This suggests that multiciliation is regulated by several redundant mechanisms to ensure the correct cell composition for the mucociliary flow. For instance, the equilibrium between multicilated and secretory cells in airway epithelia is maintained by a Jagged1–Notch signal, which emanates from Jagged-expressing multiciliated cells[43] or progenitor cells[44] and promotes both development and maintenance of secretory cells

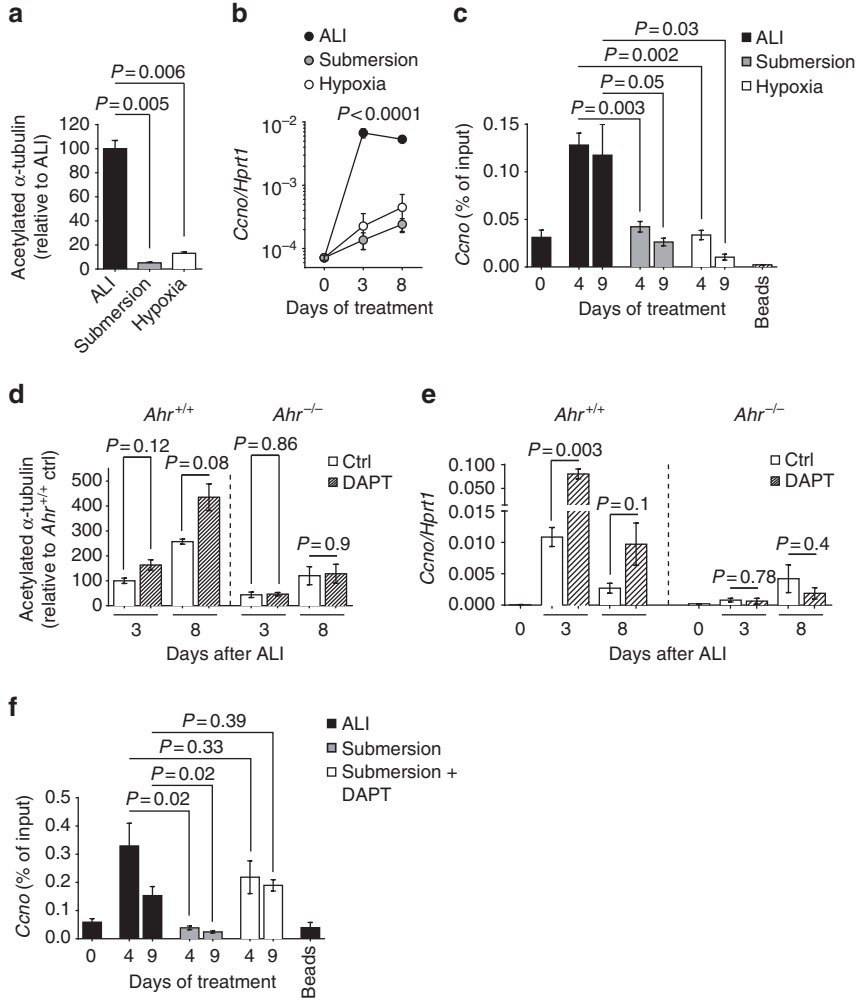

**Figure 5 | AhR-mediated *Ccno* expression is prevented by Notch signalling and requires normoxic air exposure. (a)** Quantification of acetylated α-tubulin fluorescence intensity from entire wells of AhR-sufficient mTEC cultures grown for 8 days in ALI, submersion or ALI under hypoxic condition. Fluorescence intensity of ALI condition was set as 100%. Mean ± s.e.m.; Student's *t*-test (unpaired, two-tailed). *n* = 3 wells per group, data from three independent experiments. **(b)** mRNA expression levels of *Ccno* in *Ahr*[+/+] mTEC cultures grown in ALI, submersion or ALI under hypoxic condition at indicated days of treatment. Values are normalized to *Hprt1*. Mean ± s.e.m.; two-way analysis of variance (variables: treatment and time). Data representative of three independent experiments. **(c)** Chromatin immunoprecipitation (ChIP) analysis of AhR interaction with the *Ccno* promoter in *Ahr*[+/+] mTEC cultures grown in ALI, submersion or ALI under hypoxic condition at indicated days of treatment. Mean ± s.e.m.; Student's *t*-test (unpaired, two-tailed). Data representative of three independent experiments. **(d)** Quantification of acetylated α-tubulin fluorescence intensity from entire wells of AhR-sufficient and AhR-deficient mTEC cultures at indicated days of ALI in presence or absence of DAPT. Fluorescence intensity of *Ahr*[+/+] mTEC culture at 3 days of ALI was set as 100%. Mean ± s.e.m.; Student's *t*-test (unpaired, two-tailed). Data representative of three independent experiments. **(e)** mRNA expression levels of *Ccno* in *Ahr*[+/+] and *Ahr*[−/−] mTEC cultures at indicated days of ALI in presence of DAPT or vehicle control. Values are normalized to *Hprt1*. Mean ± s.e.m.; Student's *t*-test (unpaired, two-tailed). Data representative of three independent experiments. **(f)** ChIP analysis of AhR interaction with the *Ccno* promoter in AhR-sufficient mTEC cultures grown in ALI or submersion in the presence of DAPT or vehicle control. Mean ± s.e.m.; Student's *t*-test (unpaired, two-tailed). Data representative of two independent experiments. Ctrl, control.

and inhibits development of further multiciliated cells. In the absence of multiciliated cells, Jagged1 expression is reduced, so that this tonic stimulus is missing. This might allow for increased plasticity to favour differentiation into multiciliated cells, such as transdifferentiation, where secretory cells directly differentiate into multiciliated cells when the Jagged signal is blocked or absent[45]. Transdifferentiation may be one of the back-up mechanisms that ensure the appearance of multiciliated cells in adult airway epithelia in mice deficient in CCNO or AhR. Furthermore, other transcription factors may be able to induce the *Ccno* gene in the absence of AhR in adults.

Notch and its ligands of the Dll and Jagged family are broadly expressed in the lung, and their interaction triggers signals

involved in various cell fate decisions during lung development and regeneration. Among these is the development of either multiciliated or secretory cells from basal cells in airway epithelia, where Notch2–Jagged1 interaction promotes secretory cell development and maintenance, at the expense of multiciliated cells[43,45,46]. We show here that AhR action on *Ccno* is downstream of this Notch-mediated decision, as pharmacological Notch inhibition increases ciliogenesis in wt but not in AhR-deficient epithelia.

Previous studies have reported an interplay between AhR and Notch signalling. For instance, AhR was shown to induce components of the Notch pathway such as Notch1, Notch2 and Hes in innate lymphoid cells, thus placing AhR upstream of

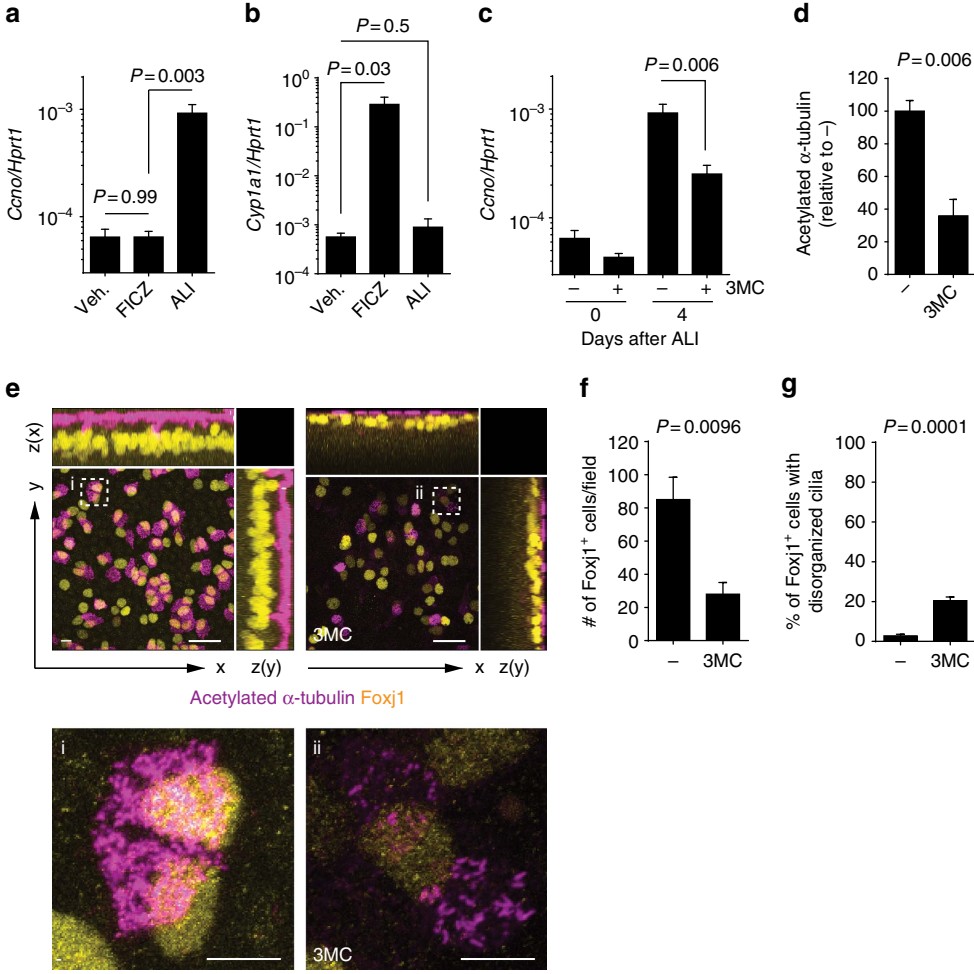

**Figure 6 | Environmental cues differentially regulate AhR transcriptional activity and can interfere with each other.** (**a,b**) mRNA expression levels of *Ccno* and *Cyp1a1* in wt mTEC cultures treated with vehicle or FICZ (for 2 days before ALI onset) or grown in ALI for 4 days. Values are normalized to *Hprt1*. Mean ± s.e.m.; Student's *t*-test (unpaired, two-tailed). Data representative of three independent experiments. (**c**) mRNA expression levels of *Ccno* in AhR-sufficient mTEC cultures at indicated days of ALI and treated with 3-methylcholanthrene (3MC) or vehicle starting 2 days before ALI onset. Values are normalized to *Hprt1*. Mean ± s.e.m.; Student's *t*-test (unpaired, two-tailed). Data representative of three independent experiments. (**d**) Quantification of acetylated α-tubulin fluorescence intensity from entire wells of $Ahr^{+/+}$ mTEC cultures at 8 days of ALI in the presence or absence of 3MC. Fluorescence intensity of vehicle-treated control ( − ) was set as 100%. Mean ± s.e.m.; Student's *t*-test (unpaired, two-tailed). Data representative of three independent experiments. (**e**) Immunofluorescent staining of acetylated α-tubulin (cilia, violet) and Foxj1 (yellow) in AhR-sufficient mTEC cultures at 8 days of ALI in the presence of 3MC or vehicle. Dashed boxes indicate insets (**i,ii**). Scale bars, 20 μm (5 μm for insets **i** and **ii**). (**f**) Number of Foxj1[+] cells per field in AhR-sufficient mTEC cultures at 8 days of ALI in the presence or absence of 3MC. Mean ± s.e.m.; Student's *t*-test (unpaired, two-tailed). $n = 5$ random fields per group. (**g**) Fraction of Foxj1[+] cells bearing a disorganized pattern of cilia (uneven distribution on the apical surface; intracytosolic staining of acetylated α-tubulin) in $Ahr^{+/+}$ mTEC cultures at 8 days of ALI in the presence or absence of 3MC. Mean ± s.e.m.; Student's *t*-test (unpaired, two-tailed). $n = 5$ random fields per group. Data in **e–g** are representative of two independent experiments.

Notch[39], whereas a study in T cells[47] proposed synergy between Notch and AhR for interleukin-22 induction. Our data suggest that in airway epithelial cells, termination of Notch signalling triggered by air exposure is required to define a permissive state for AhR binding to the *Ccno* promoter, thus promoting multiciliogenesis.

Our data suggest that AhR acts as a sensor that relays cues from the environment into targeting distinct sets of genes and thereby triggering the appropriate gene expression programme. We find that air exposure triggers AhR targeting of genes important for multiciliogenesis and toxic ligands induce detoxifying cytochromes, with no overlap in the target gene induction (Fig. 6a,b; Supplementary Fig. 7). Future studies will address how this differential targeting is achieved. For instance, it can be envisaged that AhR either associates with different partners to form different complexes for transcriptional

activation or that selective inhibition of some gene induction guides the transcriptional activity of AhR. It is interesting to note in this context that in the presence of a Notch inhibitor, AhR agonists are still unable to trigger *Ccno* induction, indicating that Notch signals are not sufficient to provide the specificity for AhR-induced ciliogenesis.

It is well known that exposure to air pollution contributes to inflammatory diseases such as bronchitis, asthma and chronic obstructive pulmonary disease[48–50]. There is also a close link between these diseases and congenic or acquired defects in mucociliary clearance. In particular, vehicle exhaust and cigarette smoke, which contain AhR ligands, are strongly associated with chronic obstructive pulmonary disease (COPD)[48]. A recent study showed that cigarette smoke reduced Foxj1 expression in human airway epithelial cultures and impaired ciliogenesis[51]. Although

AhR was not implicated in this study, smoke exposure induced *Cyp1a1* and *Cyp1b1*, which are indicators of the detoxification pathway that we have shown here interferes with the physiological role of AhR in ciliogenesis.

Our data on the effect of AhR ligands in disrupting the AhR-mediated effect on promoting ciliogenesis provide a plausible pathophysiological mechanism that explains how AhR ligands in air pollutants could contribute to respiratory diseases. Given that the influence of AhR on ciliogenesis is predominant in the early phase of development, it is conceivable that its dysregulation by environmental pollutants is particularly significant in young infants.

## Methods

**Mice.** Wild-type and AhR-deficient mice ($Ahr^{-/-}$)[19] on the C57BL/6 background were bred and maintained at the Francis Crick Institute, Mill Hill Laboratory under specific pathogen-free conditions according to the protocols approved by the UK Home Office and the local ethics committee.

**Primary mouse tracheal epithelial cell culture (mTEC).** Isolation and culture of primary mTECs were performed as previously described[18]. In brief, cells isolated by enzymatic treatment were seeded onto 0.4 μm pore size clear polyester membrane (Corning) coated with a collagen solution. Cells were grown in submersion until confluent, and then exposed to air to establish an ALI. Alternatively, cells were kept in submersion or grown at an ALI in hypoxic conditions (0.5% $O_2$) in a Ruskinn's InvivO2 hypoxia workstation. Notch pharmacological inhibition was achieved by the treatment with DAPT (40 μM, Sigma) starting at the onset of ALI. AhR activation was achieved by treating cells with the agonists FICZ (Enzo, 250 nM) or 3MC (Sigma, 1 μM) starting from day 2 before ALI.

**RNA extraction and quantitative PCR.** RNA was isolated from mTEC cultures using the Qiagen RNeasy mini kit, according to the manufacturer's instructions. One microgram total RNA was reverse-transcribed using the ThermoScript RT-PCR System kit (Invitrogen). The cDNA served as a template for the amplification of genes of interest and the housekeeping genes (*Hprt1* or *U6*) by real-time quantitative PCR, using TaqMan Gene Expression Assays (Applied Biosystems), universal PCR Master Mix (Applied Biosystems) and the ABI-PRISM 7900 sequence detection system (Applied Biosystems). mRNA expression levels were determined using the $\Delta C_t$ method, relatively to the level of *Hprt1* or *U6* gene expression.

The following probes (Applied Biosystems) have been used: *Hprt1* (Mm00446968_m1), *Ccno* (Mm01297259_m1), *Mcidas* (Mm01308202_m1), *Ccdc67* (Mm00725262_m1), *Cdc20b* (Mm01298254_m1), *Myb* (Mm00501741_m1), *Cyp1a1* (Mm00487217_m1), *Ahr* (Mm00478930), *mir449c* (001667) and *U6* (001973).

**Transfection and luciferase assay.** Primary mTECs were transiently transfected at 5 days post ALI with 400 ng of pGL4.10-CCNO vectors or pGL4.10 empty vector plus 10 ng of pRL-CMV, using Lipofectamine2000 (Invitrogen). NIH 3T3 cells plated on 24-well plates were co-transfected with 100 ng of pGL4.10-CCNO vectors or pGL4.10 empty vector together with 200 ng of the expression vector pCMV-AhR (or empty control), plus 10 ng of pRL-CMV. The Luciferase assay was performed 36 h after transfection using the Dual-Luciferase Reporter Assay Kit (Promega). Results are presented as fold induction relative to control vector pGL4.10, after normalization to Renilla luciferase activity.

**Plasmids.** For luciferase assays, the mouse locus chr13:112986313–112987887 (*Ccno* promoter) and chr13:112986313–112987887 Δ112987750–112987392 (Delta *Ccno* promoter), containing or missing (Delta) putative AhR responsive elements and encompassing the *Ccno* gene transcription start site, were cloned into the EcoRV/HindIII restriction sites of the pGL4.10 vector (Promega). pRL-CMV (Promega) was used as an internal control for transfection efficiency.

**Immunofluorescence microscopy.** Cultures were fixed in 4% paraformaldehyde, permeabilized with 0.1% Triton X-100 for 15 min at room temperature and then blocked in 0.5% bovine serum albumin for 1 h. Primary antibodies specific for acetylated α-tubulin (T7451, Sigma) and γ-tubulin (T6557, Sigma) were added apically in 0.5% bovine serum albumin and incubated for 1 h at room temperature. After washing with PBS, fluorochrome-conjugated secondary antibodies (Alexa Fluor, Life Technologies) were added for 1 h at room temperature. Finally, transwell filters were washed in PBS and mounted on slides using Vectashield mounting medium with 4,6-diamidino-2-phenylindole (Vector labs). Images were acquired either on a Zeiss LSM 710 Laser Scanning microscope (× 20 and oil immersion × 63 lenses) using the ZEN software for image analysis or on an Olympus VS120 slide scanner. Measurements of acetylated α-tubulin staining were made using regions of interest that cover the whole transwell. Binary images were generated by global thresholding and measured using the ImageJ 1.4j application.

**Scanning electron microscopy.** Trachea and mTEC samples were fixed in 2% glutaraldehyde followed by 1% osmium tetroxide in 0.1 M sodium cacodylate buffer (pH 7.2). Samples were dehydrated through a graded series of ethanols and critical point dried. Mounted samples were sputter coated with 12 nm gold. Images were acquired on a Jeol 35CF microscope.

**Chromatin immunoprecipitation.** mTECs were grown on 0.4 μm pore size clear polyester membrane six-well plates (Corning) coated with a collagen solution. At the indicated time points, $10^7$ cells were crosslinked with 1% paraformaldehyde and lysed in 1 ml of lysis buffer (1% SDS, 10 mM EDTA, 50 mM Tris–HCl, pH 8) containing protease inhibitor cocktail (Roche). Chromatin was then sheared by sonication, and centrifuged at 14,000 r.p.m. for 10 min at 4 °C; 5% of sonicated cell extract was kept as input. Supernatants were then diluted in dilution buffer (1% Triton X-100, 150 mM NaCl, 2 mM EDTA, 20 mM Tris–HCl, pH 8) and immunoprecipitated overnight at 4 °C with 2 μg of anti-AhR antibody (BML-SA210, Enzo) or 5 μg of anti-ARNT antibody (sc-8076, C-19, Santa Cruz Biotechnology). Protein G Dynabeads (Life Technologies) were then added to the cell extract for 4 h at 4 °C. Samples were washed once in low-salt buffer (0.1% SDS, 1% Triton X-100, 2 mM EDTA, 150 mM NaCl, 20 mM Tris–HCl, pH 8), once in high-salt buffer (same as low salt except for 500 mM NaCl), once in LiCl wash buffer (0.25 M LiCl, 1% NP40, 1% sodium deoxycholate, 1 mM EDTA, 10 mM Tris–HCl, pH 8) and twice with 10 mM Tris–HCl 1mM EDTA. Protein/DNA complexes were eluted in elution buffer (1% SDS, 100 mM NaHCO$_3$) at room temperature for 30 min followed by 2 min at 65 °C. Crosslinking was reversed by the addition of NaCl (200 mM final) and incubation overnight at 65 °C. After RNase and proteinase K treatment, DNA fragments were purified using the Chip DNA Clean and concentrator kit (Zymo Research) and analysed by quantitative PCR and by normalization relative to input DNA amount. The following primers were used for the *Ccno* promoter: forward 5′-GGGGCTCAGCCAGTGAGA-3′; reverse 5′-GGCGCAGCTCTAAGTACCC-3′.

**Immunoblot analysis.** Cells were lysed in RIPA buffer supplemented with protease inhibitor cocktail (Roche). Total cell extracts were run on a poly-acrylamide gel and transferred to the nitrocellulose membrane (Whatman). The following primary antibodies were used: anti-AhR antibody (BML-SA210, Enzo, 1:1,000) and anti-β-actin (4970s, Cell Signaling Technology, 13E5, 1:5,000). The immunoblot was developed using horseradish peroxidase-conjugate secondary antibodies (Bio-Rad) and the ECL Western Blotting Detection kit, as instructed by the manufacturer (GE Healthcare). Full scans in Supplementary Fig. 3.

***X. laevis* experiments.** *X. laevis* embryos were obtained by *in vitro* fertilization and staged according to Nieuwkoop and Faber[52]. An amount of 5 ng each of morpholino (purchased from Gene Tools LLC, Philomath, OR, USA) against either *X. laevis* Ahr1α and β (MO *Ahr1*α 5′-ATGATGTTCGTGTTCATCCTACTCC-3′, MO *Ahr1*β 5′-TATGATGTTCGTGTTCATTCTGCTC-3′) or 5 ng mismatch controls (control *Ahr1*α 5′-ATCATGTTAGTCTTCATCATACTAC-3′, control *Ahr1*β 5′-TATCATATTCGTCTTAA TTCTGATC-3′) were co-injected with 50 pg of *Xlt* Centrin2-GFP mRNA[53] (obtained from John Wallingford via The European Xenopus Stock Centre) into the two ventral blastomeres at the four-cell stage. Embryos were fixed at stage 26–28 in MEMFA for 1 h, permeabilized in PBDT (PBS, 0.5% Triton X-100, 1% dimethylsulphoxide) for 30 min, and then incubated in blocking solution (PBDT + 1% goat serum) for 1 h. Acetylated α-tubulin was detected using 1:2,000 dilution of a mouse monoclonal antibody (Sigma T7451) in blocking solution overnight at 4 °C. Embryos were then washed five times for 1 h, followed by an overnight incubation at 4 °C with 1:1,000 dilution of goat anti-mouse Alexa568 labelled IgG (A11031, Life Technologies). The next day embryos were washed five times for 1 h, then mounted onto glass coverslips using Prolong Gold antifade mounting medium (ThermoFisher Scientific). Embryos were imaged with a Zeiss LSM 710 confocal microscope using × 63 oil immersion lens. Image analysis was done using either ImageJ, ZEN blue or ZEN black software packages. Quantification of acetylated α-tubulin staining per cell was determined by selecting a region of interest manually and then determining the background corrected integrated density of fluorescence using ImageJ. Over a 100 cells from six embryos were imaged per condition.

**Statistical analyses.** Comparisons between two groups were assessed applying the Student's *t*-test (unpaired, two-tailed). Comparisons between more than two groups were assessed applying the two-way analysis of variance (with Sidak's multiple comparison test in Fig. 3b) and the variables of the analysis indicated in the figure legends. Statistical significance is indicated as precise *P* values except for significance lower than 0.0001 or > 0.99. All statistical analyses were performed using Prism 6 software (Graphpad).

**Data availability.** We declare that the data supporting the findings of this study are available within the article and from the authors on request.

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

## Acknowledgements

This work was supported by the Francis Crick Institute which receives its core funding from Cancer Research UK (FC001206, FC001159, FC001157), the UK Medical Research Council (FC001206, FC001159, FC001157) and the Wellcome Trust (FC001206, FC001159, FC001157). In addition, this research was supported by MRC grant U117597139 (S.C. and A.W.), a Wellcome Investigator Grant to B.S. and a Boehringer Ingelheim Fonds PhD Fellowship grant to M.V. We would like to acknowledge support by the Biological Research Facility for breeding and maintenance of our animals, as well as expert help by the Light Microscopy Facility, the Advanced Sequencing Facility and the Electron Microscopy Facility.

## Author contributions

M.V., S.C., K.S.D. and E.M.A.H. performed the experiments. M.V., S.C., K.S.D., E.M.A.H., M.G., H.A., J.C.M., B.S. and A.W. analysed and interpreted the data. M.V., S.C., B.S. and A.W. wrote the manuscript. K.S.D. and J.C.M. contributed to manuscript writing.

## Additional information

**Competing financial interests:** The authors declare no competing financial interests.

