## [Peer Review File · Nature Communications]

Reviewers' comments:

Reviewer #1 (Remarks to the Author):

The manuscript by Villa et al examines the role of AhR in multiciliated cell differentiation. They show that multiciliated cell differentiation is impaired in AhR^{-/-} mouse trachea, and in cultured primary tracheal cells (mTEC) from AhR mutant animals. They also show that AhR knock down using morpholinos disrupts multiciliated cell formation in the *Xenopus* larval skin. They provide expression data from mTECs that key regulators of multiciliated cell differentiation are downregulated in the absence of AhR, including *Ccno*, *Mcidas*, *Myb* and *Ccdc67*. They use ChIP to show that AhR binds to the *Ccno* promoter during differentiation and provide evidence that AhR acts downstream of Notch, since Notch inhibition cannot overcome a loss of AhR activity. Finally, they show that agonists of AhR inhibit differentiation in mTECs, although mechanistically this is not examined in detail. The identification of AhR as a novel player required for multiciliated cell differentiation in diverse systems is a contribution, particularly in light of the possibility that activation of this receptor by environmental pollutions could influence this cell type in the lung. The authors, however, need to clarify a number of issues concerning the data, as listed below.

1) Figure 1: In panel D, a cell is circled in the mutant culture, but it is not clear how this cell was chosen. Is it a multiciliated cell, or just another cell type that is known to form in these cultures?

The reason for quantifying the phenotype using the two approaches in panel b and c is not very clear. What is the difference between the images shown in Figure 1a and Extended Fig. 1a? The scale bar for Extended Fig.1a indicates 1mm which seems such low power that the results not very useful.

The implication is that the loss of AnR leads to two phenotypes: (1) a reduction in the number of multiciliated cells, and (2) defective multiciliated cell differentiation where basal bodies form, but have trouble docking and extending cilia. A clearer scoring paradigm, perhaps using additional antibodies (e.g. pericentrin), would better document the two phenotypes.

2) Figure 2: The images used in panel b to illustrate how the data were obtained are too small to see the data, while there is a large blank area below in the Figure where they could be increased in size.

To compare the frog data to that obtained in the mouse, the authors need to score whether the number of multiciliated cells change (compared to other cell types in the skin) and what percentage of the multiciliated cells that do form show arrested differentiation.

Morpholino specificity is not well-controlled. The implication is that both isoforms of AhR need to be blocked to get a phenotype. Showing in extended data that blocking either one alone has no effect would provide some evidence for specificity. Finally, I may have missed this but is there evidence that both forms are expressed in the skin of early *Xenopus* embryos? Does this explain why both forms need to be inhibited in *Xenopus* but not in mouse?

3) Figure 3: The data here seems solid, but the focus on *Ccno* using ChIP for AhR is not explained. Why did the authors choose *Ccno*? Have the authors examined binding to promoters of other genes that are known to be key regulators, such as *Mcidas* or *Myb*? What is the negative control or is AhR binding to all genes that are upregulated during multiciliated cells differentiation? This latter data is presumably why *Ccno* figures prominently in the Title, but the evidence that *Ccno* is the most critical target is not clear.

Along the same lines, the role of *Foxj1* is largely ignored in the analysis, even though the phenotype presented resembles in many ways the one seen when *Foxj1* is eliminated (basal body docking defects and poor cilia extension). As commercial *Foxj1* antibodies are readily available for the mouse protein, it is not clear why the authors haven't examined how its expression is altered in the AhR mutant.

4) Figure 4: Perhaps the most interesting part of the manuscript is that when AhR is activated using an agonist, multiciliated cell differentiation is impaired. This observation could be substantiated in several ways. Does the agonist interfere with multiciliated cell differentiation in *Xenopus*? This would provide evidence the mechanism is conserved. Does AhR binding to the *Ccno* promoter decrease upon treatment with the agonist? How does the agonist affect multiciliated cell differentiation? The images shown in extended data Figure 4B are extremely poor quality, and it would be better to stain with Hoescht and gamma tubulin, scoring the number of cells that differentiated, as well as the number with arrested differentiation.

5) The role of Notch in restoring differentiation in submerged conditions has been published. The extended data in Figure 3 is not very compelling given the staining shown in the images, and so it is not clear that this adds much to the manuscript.

Reviewer #2 (Remarks to the Author):

This manuscript describes a newly discovered function for the Ah receptor in epithelial multiciliogenesis. The results provided appear to be well controlled and nicely presented. The manuscript is very well written and would be of significant interest to a number of fields. However, there are some concerns about the level of experimental support for the proposed

mechanism as detailed below.

Specific comments:

The authors propose that the mechanism of AHR mediated development is through enhanced expression of *Ccno*, which would contribute to the development of cilia. However, the only direct evidence provided is a ChIP assay, which only provides evidence that the AHR is directly or indirectly bound to the *Ccno* promoter. Additional evidence that the AHR directly drives expression of *Ccno* is needed. The addition of gel shift assays and promoter reporter assays would greatly strengthen the case that the AHR can directly enhance *Ccno* expression.

If the AHR is directly regulating *Ccno* expression, the use of an AHR antagonist would further illustrate that the AHR is specifically present on the *Ccno* promoter.

The data provided to address the mechanism of AHR activity is all based on mRNA analysis, the actual level of CCNO protein levels in wild-type versus AHR knockout cells should be provided.

On line 145 and 148 the term "classical" is used to discussing agonist, and is not defined what this actually means. It would appear that it would be better to state full agonist or potent agonist or just agonist.

Reviewer #3 (Remarks to the Author):

This paper by Villa et al. describes a series of experiments designed to address the function of AhR in developing lung epithelium. The data in this paper and the model they support are very interesting. I believe the discovery being described is of general interest and the experiments in *Xenopus* add a very nice angle to the story. Despite this, the manuscript is somewhat thin on mechanism as written. The importance of AhR in direct regulation of *Ccno* gene transcription is striking but so many questions are left unanswered. These include:

- 1) Does AhR also directly regulate the other genes implicated in multiciliogenesis as studied in the paper?
 - 2) What is the dimerization partner of AhR at the *Ccno* promoter?
 - 3) Is the effect of Hypoxia and submersion mediated by titration of Hif1 β away from AhR?
 - 4) To what extent is the studied effect mediated by a cell fate switch, or to transdifferentiation?
- This is particularly relevant given the importance of Jagged expressed within multiciliated cells

in suppression of multiciliated cell development of neighbours (Zhang et al., *Developmental Dynamics* 242:678-686, 2013) and the ongoing requirement for Jagged expression in maintenance of the non-multiciliated cell fate (Lafkas et al., *Nature*. 528. 127-131, 2015). This issue may also be relevant to redundancy in the system.

5) At what level does the redundancy operate? Is there a different bHLH-PAS protein complex that binds to the *Ccno* promoter in adults?

6) Is the phenotype quantitative (within cells) or quantitative with respect to the number of cells that are multiciliated. This is important but not very clear.

7) What happens in the presence of FICZ and DAPT? Is inhibition of Notch signalling sufficient to induce *Ccno* and other genes involved in multiciliogenesis when FICZ is present.

In addition, the manuscript is based on duplicate experiments in almost every case, it would be improved significantly if experiments were performed three times. This is particularly important given that the phenotype is transient. Also, many of the experiments seem to be distinct when this would not be expected to be the case. For example, are the samples analyzed in figure 3e different from those analyzed in Figure 3a and 3c? if so, why? If not, the Figure legend needs clarification.

Some of the sentences in this manuscript are awkward and the manuscript could be improved with some editing (eg. " The development, maintenance, and repair of epithelial tissues all need to be tightly controlled if their functions are to be maintained." This sentence doesn't really make very much sense. And "Such cells can be identified by staining for acetylated α -tubulin, and we discovered a significant reduction in the number of multiciliated cells in mTEC cultures derived from AhR deficient mice after 9 days exposure to air (Fig. 1a, b), as well as in the overall level of acetylated α -tubulin (Fig.1c, Extended Data Fig. 1a), compare to wild-type mTECs." This is a run on sentence that is very difficult to follow.").

Reviewers' comments and our replies:

Reviewer #1 (Remarks to the Author):

The manuscript by Villa et al examines the role of AhR in multiciliated cell differentiation. They show that multiciliated cell differentiation is impaired in AhR^{-/-} mouse trachea, and in cultured primary tracheal cells (mTEC) from AhR mutant animals. They also show that AhR knock down using morpholinos disrupts multiciliated cell formation in the Xenopus larval skin. They provide expression data from mTECs that key regulators of multiciliated cell differentiation are downregulated in the absence of AhR, including Ccno, Mcidas, Myb and Ccdc67. They use ChIP to show that AhR binds to the Ccno promoter during differentiation and provide evidence that AhR acts downstream of Notch, since Notch inhibition cannot overcome a loss of AhR activity. Finally, they show that agonists of AhR inhibit differentiation in mTECs, although mechanistically this is not examined in detail.

The identification of AhR as a novel player required for multiciliated cell differentiation in diverse systems is a contribution, particularly in light of the possibility that activation of this receptor by environmental pollutions could influence this cell type in the lung.

The authors, however, need to clarify a number of issues concerning the data, as listed below.

1) Figure 1: *In panel D, a cell is circled in the mutant culture, but it is not clear how this cell was chosen. Is it a multiciliated cell, or just another cell type that is known to form in these cultures?*

Our reply: As no other ciliated cell type is present in the airways and in the cultures, the cells circled in figure 1d are both of the multiciliated cell lineage. On the surface of *Ahr*^{+/+} cells, cilia are numerous, well spread out and cover the entirety of the cell surface. On *Ahr*^{-/-} cells, the filamentous structures are fewer, only cover focal areas of the apical surface and appear shorter and disorganized. Figure 1e shows a similar phenotypic difference in confocal microscopy with acetylated α -tubulin staining for the cilia, and the new figure 2 shows the same ciliar phenotypes in Foxj1-positive cells (see new figure 2a inset i – wild-type – as compared to inset ii – *Ahr*^{-/-}), confirming that such cells belong to the multiciliated lineage.

The reason for quantifying the phenotype using the two approaches in panel b and c is not very clear. What is the difference between the images shown in Figure 1a and Extended Fig. 1a? The scale bar for Extended Fig.1a indicates 1mm which seems such low power that the results not very useful.

Our reply: The measurements are complementary as slightly different parameters are assessed: figure 1b shows the quantification of acetylated α -tubulin⁺ cells per random field (5 random fields quantified) and therefore indicates the number of multiciliated cells. Figure 1c shows the quantification of total acetylated α -tubulin signal in the entire well, which allows for global unbiased quantification. Therefore, in fig. 1a, representative fields, as used for fig. 1b, are depicted, while in supplementary fig. 1a, images of entire culture wells - used for the quantification in fig. 1c - at low magnification are shown. Please note that all cultures are confluent and therefore the transwell membranes are completely covered with cells, as also

confirmed by DAPI staining (not shown). This is now better specified in the manuscript text on page 4, line 75-79. We have improved the image quality of Supplementary Figs 1a, 3a and 4b to better show the acetylated α -tubulin staining pattern. These images will remain at low magnification as they were used for global measurement.

The implication is that the loss of AhR leads to two phenotypes: (1) a reduction in the number of multiciliated cells, and (2) defective multiciliated cell differentiation where basal bodies form, but have trouble docking and extending cilia. A clearer scoring paradigm, perhaps using additional antibodies (e.g. pericentrin), would better document the two phenotypes.

Our reply: This is correct and we are grateful to Reviewer #1 for pointing out that the dual nature of the phenotype could be demonstrated better. We performed immunofluorescent staining to test Foxj1 and acetylated α -tubulin expression in *Ahr*^{+/+} and *Ahr*^{-/-} mTEC cultures. The new figure 2 shows that *Ahr*^{-/-} mTEC cultures have fewer Foxj1⁺ cells as compared to AhR sufficient mTEC cultures. In addition, a large fraction of Foxj1⁺ cells in the *Ahr*^{-/-} mTEC cultures have a disorganized ciliar pattern, characterized by few or no cilia present on the apical surface and acetylated α -tubulin staining that extends to the cytosol or along the basolateral membrane (text on page 5, line 89-100). These data extend the data shown in figure 1 and indicate that AhR deficiency influences both cell fate choice (fewer Foxj1⁺ cells) and completion of the differentiation programme (disorganized cilia pattern).

2) Figure 2: The images used in panel b to illustrate how the data were obtained are too small to see the data, while there is a large blank area below in the Figure where they could be increased in size.

Our reply: Figure 2b (now figure 3b) has been rearranged to increase the size of the panels that illustrate how cells were categorized.

To compare the frog data to that obtained in the mouse, the authors need to score whether the number of multiciliated cells change (compared to other cell types in the skin) and what percentage of the multiciliated cells that do form show arrested differentiation.

Our reply: We have now scored the fraction of cells committed to the multiciliated cell lineage (new figure 3e, scoring cells containing acetylated α -tubulin in the cell body) and find that in *Xenopus*, AhR knockdown results in disorganized cilia (fig. 3c, d), but not in numeric reduction of cells of the multiciliated lineage (fig. 3e). This is now documented in the text on page 6, line 119-123). Whether the difference between mouse and frog reflects the different techniques of reducing/deleting AhR expression or represents true differences in the biology of multiciliated cells remains to be determined.

Morpholino specificity is not well-controlled. The implication is that both isoforms of AhR need to be blocked to get a phenotype. Showing in extended data that blocking either one alone has no effect would provide some evidence for specificity. Finally, I may have missed this but is there evidence that both forms are expressed in the skin of early Xenopus embryos? Does this explain why both forms need to be inhibited in Xenopus but not in mouse?

Our reply: *Xenopus laevis* is pseudotetraploid and expresses two AhR paralogues (AhR1 α and β), both of which are expressed in *Xenopus* epidermis (Lavine JA, Toxicol. Sci. 2005 (88),

60-72). We show below that depletion of both paralogues is required to generate defects in multiciliogenesis.

Morpholinos targeting either *Ahr1 α* or *Ahr1 β* are not sufficient to impair ciliogenesis in *Xenopus laevis* embryos. (a-c) Immunofluorescent staining of acetylated α -tubulin (cilia, cyan) in the skin of embryonic *Xenopus laevis*, injected with either *Ahr1 α* MO or *Ahr1 β* MO or both. Injected cells are centrin 2⁺ (centrin 2 localizes at the basal bodies, yellow). Arrowheads in (c) indicate examples of cells with disorganized ciliary pattern.

3) Figure 3: The data here seems solid, but the focus on *Ccno* using ChIP for AhR is not explained. Why did the authors choose *Ccno*? Have the authors examined binding to promoters of other genes that are known to be key regulators, such as *Mcidas* or *Myb*? What is the negative control or is AhR binding to all genes that are upregulated during

multiciliated cells differentiation? This latter data is presumably why Ccno figures prominently in the Title, but the evidence that Ccno is the most critical target is not clear.

Our reply: The decision of focusing on *Ccno* comes from a series of recently published findings that showed *Ccno* to be central in the generation of multiple motile cilia (Wallmeier J, Nat. Genetics 2014; Funk MC, EMBO J. 2015). These data showed that absence of *Ccno*, both in mice and humans, leads to failure of basal bodies to dock at the apical membrane and impaired cilia formation. The phenotype of our *Ahr*^{-/-} mTEC cultures led us to hypothesize that *Ccno* may be controlled by AhR.

As proposed by the Reviewer, we have now explored the *Mcidas* promoter to test for AhR binding. We have designed five primer pairs spanning the entire 3 kb region between *Ccno* and *Mcidas* (new Supplementary Fig. 2d) and used these for ChIP-PCR analysis. We did not find evidence for direct binding of AhR to the promoter of *Mcidas* (new Supplementary Fig. 2f, text on page 7, line 149-150). However, *Mcidas* is positioned only 3 kb downstream of the *Ccno* gene (Supplementary Fig. 2a, d), and we therefore suggest that the binding of AhR in the *Ccno* promoter region may drive expression of *Mcidas* due to its close vicinity. In addition, these data provide proof that AhR does not bind indiscriminately to any gene upregulated during multiciliogenesis.

Along the same lines, the role of Foxj1 is largely ignored in the analysis, even though the phenotype presented resembles in many ways the one seen when Foxj1 is eliminated (basal body docking defects and poor cilia extension). As commercial Foxj1 antibodies are readily available for the mouse protein, it is not clear why the authors haven't examined how its expression is altered in the AhR mutant.

Our reply: We agree that Foxj1 is an important factor in multiciliogenesis and have in fact used Foxj1 staining to address the request of Reviewer #1 to better define the dual nature of the mouse phenotype, see above under point 1 and in the new figure 2 of the study. The text describing these findings is on page 5, line 89-100.

4) Figure 4: *Perhaps the most interesting part of the manuscript is that when AhR is activated using an agonist, multiciliated cell differentiation is impaired. This observation could be substantiated in several ways. Does the agonist interfere with multiciliated cell differentiation in Xenopus? This would provide evidence the mechanism is conserved. Does AhR binding to the Ccno promoter decrease upon treatment with the agonist? How does the agonist affect multiciliated cell differentiation? The images shown in extended data Figure 4B are extremely poor quality, and it would be better to stain with Hoescht and gamma tubulin, scoring the number of cells that differentiated, as well as the number with arrested differentiation.*

Our reply: We have now performed experiments in *Xenopus* using the strong AhR agonists PCB126 (1 μ M) and FICZ (100 nM) and find that multiciliogenesis is not affected under these conditions (figure below).

Strong AhR agonists affect neither multiciliated cell development nor ciliogenesis in *Xenopus laevis* embryos. Immunofluorescent staining of acetylated α -tubulin (cilia, red) and by phalloidin (cell membrane, cyan) in the skin of embryonic *Xenopus laevis*, treated with the AhR agonists FICZ (100 nM) or PCB126 (1 μ M).

We have also measured AhR binding to the *Ccno* promoter upon 3MC exposure. In contrast to the highly sensitive and robust quantitative assays measuring *Ccno* expression and acetylated α -tubulin signal (Fig. 4i, j, now 5i, j) which show a significant and reproducible three-fold reduction, it is more difficult to obtain convincing ChIP-PCR data that show differences in that order of magnitude.

As explained above under point 1, we have now improved the quality of images showing the entire well, and we still think this is the most global and unbiased way of quantifying differences between cultures. We show in the new figure 6 that treatment of AhR sufficient mTECs with strong, poorly metabolised agonists leads to both reduced numbers of cells in the multiciliated lineage (Fig. 6e, f) and to disorganised cilia in the remaining multiciliated cells (Fig. 6e, g), similar to what is seen in AhR deficiency. The text referring to new data in Fig. 6 is on page 9, line 196-204.

5) The role of Notch in restoring differentiation in submerged conditions has been published. The extended data in Figure 3 is not very compelling given the staining shown in the images, and so it is not clear that this adds much to the manuscript.

Our reply: We agree that the role of Notch in restoring differentiation under conditions of submersion is established, but in our view Supplementary Fig. 3 is useful to illustrate the biology underlying the experiment shown in Fig. 4f (now 5f). We have improved the quality of Supplementary Fig. 3a used for global analysis.

Reviewer #2 (Remarks to the Author):

This manuscript describes a newly discovered function for the Ah receptor in epithelial multiciliogenesis.

The results provided appear to be well controlled and nicely presented. The manuscript is very well written and would be of significant interest to a number of fields.

Our reply: We thank the Reviewer for the appreciation of our study.

However, there are some concerns about the level of experimental support for the proposed mechanism as detailed below.

Specific comments:

The authors propose that the mechanism of AHR mediated development is through enhanced expression of *Ccno*, which would contribute to the development of cilia. However, the only direct evidence provided is a ChIP assay, which only provides evidence that the AHR is directly or indirectly bound to the *Ccno* promoter. Additional evidence that the AHR directly drives expression of *Ccno* is needed. The addition of gel shift assays and promoter reporter assays would greatly strengthen the case that the AHR can directly enhance *Ccno* expression.

Our reply: We now show in a new Fig. 4e a luciferase reporter assay in *Ahr*^{+/+} and *Ahr*^{-/-} mTECs using constructs either spanning a 1.5 kb region upstream of the *Ccno* gene (*Ccno* promoter), or the same region without putative AhR response elements (Delta *Ccno* promoter). Only the construct containing the AhR binding sites led to reporter gene activation, and only in mTECs expressing AhR (fig. 4e, text on page 7, line 139-144). This indicates that indeed AhR is driving gene expression from the *Ccno* promoter.

We confirmed this data more directly by co-transfecting NIH3T3 cells with a vector encoding AhR and a vector carrying a luciferase gene under the control of either the 1.5 kb promoter region or the Delta version. As shown below, AhR drives reporter gene expression from the *Ccno* promoter, and this requires AhR-specific response elements.

AhR binds *Ccno* promoter and drives gene transcription. Luciferase reporter assay to quantify the potential of AhR to drive gene expression was performed in NIH3T3 cells co-transfected with a plasmid encoding AhR (pCMV-*Ahr*) or empty vector (pCMV) and a luciferase-encoding reporter gene downstream of the *Ccno* promoter. Cells were analysed 36 hours post-transfection. Delta *Ccno* promoter lacks predicted AhR response elements. Mean \pm SEM.

If the AHR is directly regulating *Ccno* expression, the use of an AHR antagonist would further illustrate that the AHR is specifically present on the *Ccno* promoter.

Our reply: We tried this experiment using CH223191 and did not find any change in *Ccno*

expression. This in line with our findings that a different set of rules governs AhR binding to the *Ccno* promoter as compared to inducing detoxifying genes such as *Cyp1a1*. It has also been shown in previous papers that CH223191 is able to antagonise some but not all AhR ligands (Zhao B, Tox. Sci. 2010). We are currently exploring the molecular mechanisms that drive the differential ability of AhR to target *Cyp1a1* and *Ccno*, and it is conceivable that an antagonist such as CH223191 may be effective in inhibiting AhR-dependent *Cyp1a1* activation but not *Ccno*-activation.

The data provided to address the mechanism of AHR activity is all based on mRNA analysis, the actual level of CCNO protein levels in wild-type versus AHR knockout cells should be provided.

Our reply: We agree and have tested a number of commercially available α CCNO mAbs, which unfortunately only produced a series of unspecific bands on our Western blots or a very faint staining in IF, thus not allowing unequivocal interpretation of the data. Two commercial suppliers of the mAb used in previously published studies (e.g. Wallmeier, Nat. Genetics 2014 (46), 646-651) communicated to us that the mAb is presently not available.

On line 145 and 148 the term "classical" is used to discussing agonist, and is not defined what this actually means. It would appear that it would be better to state full agonist or potent agonist or just agonist.

Our reply: We have now changed this to “strong” or “potent” agonist.

Reviewer #3 (Remarks to the Author):

This paper by Villa et al. describes a series of experiments designed to address the function of AhR in developing lung epithelium.

The data in this paper and the model they support are very interesting. I believe the discovery being described is of general interest and the experiments in Xenopus add a very nice angle to the story.

Our reply: We thank the reviewer for the positive comments.

Despite this, the manuscript is somewhat thin on mechanism as written. The importance of AhR in direct regulation of Ccno gene transcription is striking but so many questions are left unanswered. These include:

1) Does AhR also directly regulate the other genes implicated in multiciliogenesis as studied in the paper?

Our reply: As we explained above, under point 3 of Reviewer #1, we explored AhR binding to the *Mcidas* promoter, and using five different primer pairs, we did not find evidence for AhR directly targeting the *Mcidas* gene. This data is now shown in (Supplementary Fig. 2d,f, text on page 7, line 149-150). We agree with the reviewer that a more global study of the AhR targets in multiciliogenesis is warranted and we will attempt this in the future, however this is challenging with a transwell-based primary cell culture system where scaling up cell

numbers is not straight forward, and we feel that this approach would be beyond the scope of the current study.

2) What is the dimerization partner of AhR at the *Ccno* promoter?

3) Is the effect of Hypoxia and submersion mediated by titration of Hif1 β away from AhR?

Our reply: In order to answer these questions, we have performed ChIP PCR using anti-ARNT mAbs. We find an increase of ARNT binding to the *Ccno* promoter upon air exposure (Supplementary Fig. 2e, text on page 7, line 144-149), closely reflecting the increased *Ccno* targeting by AhR (now Fig. 4d) and strongly suggesting that the generally acknowledged AhR binding partner ARNT is involved in *Ccno* targeting by AhR. This is also supported by co-immunoprecipitation data experiments showing direct association of AhR and ARNT in airway epithelia upon air exposure (figure below). Since air exposure increases AhR expression (Fig. 4b), more material is pulled down in this situation as opposed to submersion and hypoxia, which makes comparisons difficult to interpret (a, below). We therefore present this data here but did not include it in the paper. In an attempt to corroborate the data presented below in (a) we performed ChIP PCR using anti ARNT mAbs on AhR sufficient mTEC cultures grown in ALI, submersion and hypoxia (b, below). ARNT binds the *Ccno* promoter upon air exposure, however the binding is prevented by submersion or hypoxia. The reduced availability of ARNT at the *Ccno* promoter may reflect the involvement of ARNT in other cellular pathways (e.g. dimerization with HIF1 α in hypoxic conditions) or a reduction in the representation of ARNT in the protein pool.

This data suggests that ARNT is the dimerization partner of AhR on *Ccno* promoter and that submersion or hypoxic conditions affect the ability of both AhR and ARNT to bind the *Ccno* locus.

Conditions of submersion and hypoxia titrate ARNT away from AhR and dampen its binding to *Ccno* promoter. (a) Immunoblot analysis of AhR, ARNT and β -actin protein expression in AhR sufficient mTEC cultures grown for 5 days in ALI, submersion or ALI under hypoxic condition (0.5% O₂). Samples have been immunoprecipitated using an α -AhR mAb (IP: α -AhR) to assess binding of ARNT to AhR in the indicated conditions. (b) ChIP analysis of ARNT interaction with the *Ccno* promoter in AhR sufficient mTEC cultures grown for 5 days in ALI, submersion or ALI under hypoxic condition. Mean \pm SEM; Student's *t* test (unpaired, two-tailed).

4) *To what extent is the studied effect mediated by a cell fate switch, or to transdifferentiation? This is particularly relevant given the importance of Jagged expressed within multiciliated cells in suppression of multiciliated cell development of neighbours (Zhang et al., *Developmental Dynamics* 242:678-686, 2013) and the ongoing requirement for Jagged expression in maintenance of the non-multiciliated cell fate (Lafkas et al., *Nature*. 528. 127-131, 2015). This issue may also be relevant to redundancy in the system.*

Our reply: We are grateful to Reviewer #3 for bringing to our attention transdifferentiation as a possible mechanism of redundancy, which could explain the mild or absent phenotype in adult airways found here and in other studies such as, for instance, in the *Ccno* deficient mouse model. It is conceivable that the absence of multiciliated cells may reduce the tonic Jagged-driven signal for maintenance of secretory cell fate and for suppression of multiciliated cell fate. This may allow unconventional differentiation pathways from fully differentiated secretory cells to multiciliated cells and may explain how in the absence of factors such as AhR or *Ccno*, multiciliated cells that are strongly compromised around birth nevertheless eventually differentiate in adults. We have added this intriguing possibility to the discussion on page 10-11, line 228-244.

5) *At what level does the redundancy operate? Is there a different bHLH-PAS protein complex that binds to the *Ccno* promoter in adults?*

Our reply: We agree with the referee that this is a fascinating question, which we will address in the future. This question is now included in the discussion page 11, line 244-245.

6) *Is the phenotype quantitative (within cells) or quantitative with respect to the number of cells that are multiciliated. This is important but not very clear.*

Our reply: We have replied to a similar request from Reviewer #1 and have included a new figure (Fig. 2) to address this important question: We performed immunofluorescent staining to test Foxj1 and acetylated α -tubulin expression in *Ahr*^{+/+} and *Ahr*^{-/-} mTEC. Figure 2 shows that *Ahr*^{-/-} cultures have fewer Foxj1⁺ cells as compared to AhR sufficient mTEC. Further confirming the data shown in figure 1, a large fraction of Foxj1⁺ cells from *Ahr*^{-/-} mTEC have a disorganized ciliary pattern, characterized by few or no cilia present on the apical surface and acetylated α -tubulin staining that extends to the cytosol or along the basolateral membrane. These data suggest that AhR deficiency impacts both at the level of cell fate decision (fewer Foxj1⁺ cells) and at the structural level (disorganized cilia pattern in Foxj1⁺ multiciliated cells). The text explaining these new data is on page 5, line 89-100.

7) *What happens in the presence of FICZ and DAPT? Is inhibition of Notch signalling sufficient to induce *Ccno* and other genes involved in multiciliogenesis when FICZ is present.*

Our reply: As shown below, inhibition of Notch signalling is not sufficient to induce *Ccno* when FICZ or another AhR agonist, 3MC, are present. We show that both in submersion (a, b), and with air exposure (c), DAPT addition to agonist-treated cultures does not significantly change *Ccno* expression relative to agonist-only treated cells. Moreover, even when the

Notch-mediated block of *Ccno* induction is relieved (DAPT), AhR agonists are still not able to induce *Ccno* (compare DAPT vs DAPT+agonist). Therefore, the inability of AhR agonists to induce *Ccno* is not merely due to a Notch-mediated blockade.

In the presence of FICZ and 3MC, Notch blockade is unable to induce *Ccno* expression. a-c, mRNA expression levels of *Ccno* were measured in AhR sufficient mTEC cultures at 0 and 4 days after onset of ALI or submersion. Cells were treated with the AhR agonists FICZ or 3MC and the γ -secretase inhibitor DAPT as indicated. Values are normalized to *Hprt1*. Mean \pm s.e.m. One-way ANOVA + Tukey's multiple comparisons test.

In addition, the manuscript is based on duplicate experiments in almost every case, it would be improved significantly if experiments were performed three times. This is particularly important given that the phenotype is transient. Also, many of the experiments seem to be distinct when this would not be expected to be the case. For example, are the samples analyzed in figure 3e different from those analyzed in Figure 3a and 3c? if so, why? If not, the Figure legend needs clarification.

Our reply: This was a mistake as in fact, the vast majority of experiments have been done more than twice. The figure legends have now been amended to reflect this.

All the experiments shown in Fig. 4 (old Fig. 3) come now from the same samples.

Some of the sentences in this manuscript are awkward and the manuscript could be improved

with some editing (eg. " The development, maintenance, and repair of epithelial tissues all need to be tightly controlled if their functions are to be maintained." This sentence doesn't really make very much sense. And "Such cells can be identified by staining for acetylated α -tubulin, and we discovered a significant reduction in the number of multiciliated cells in mTEC cultures derived from AhR deficient mice after 9 days exposure to air (Fig. 1a, b), as well as in the overall level of acetylated α -tubulin (Fig.1c, Extended Data Fig. 1a), compare to wild-type mTECs." This is a run on sentence that is very difficult to follow.").

Our reply: We thank the reviewer for pointing out these shortcomings. As we now comply with the Nature Communications format of short Abstract and separate Introduction, Results and Discussion sections, we have modified many passages and in particular have removed or broken up the above phrases into shorter, more comprehensible ones.

Reviewers' Comments:

Reviewer #1 (Remarks to the Author)

The manuscript by Villa et al has improved through additional analyses and clarification. My only comment is that the field of multiciliated cell transcription is rapidly evolving, with the addition of *Gemc1* and *Tp73* as important players. It is not necessary that the authors address these new players in the results, but they should be at least referenced and incorporated, as part of the Discussion. In addition, the first reported example of a transition phenotype in the mouse was that described in the *Myb* mutants, and this needs to be referenced on lines 107.

Reviewer #2 (Remarks to the Author)

This article breaks new ground in understanding the role of the AHR in various cellular processes. The authors have added a considerable amount of new data where possible to more firmly support the conclusions and thus the manuscript is significantly improved. The authors have meaningfully address most of the key comments of the reviewers.

Reviewer #3 (Remarks to the Author)

As this review is for a resubmission, I will forgo summarizing the manuscript. It is, however, important to note that the finding in this manuscript are novel and of general interest. The manuscript has been improved substantially since first submission. For example, some of the new figures are clearer. Also, the phenotype is described in greater detail with respect to aspects associated with cell fate (in the mouse) and to morphogenesis or fidelity of differentiation (in the mouse and in *Xenopus*). The new experiments and the new text really improve the paper. For example, the new text on page 5 helps to clarify the phenotype and the new text on page 7 establishes a role for ARNT. These are nice additions to the story. Indeed, all of the new data has improved the story substantially and the paper is now both exciting and logically presented. The discussion is very well done and puts the work in context. The rebuttal letter is both respectful and comprehensive. Indeed, the experiments in the rebuttal letter should be included in the supplementary data as they really add to the story.

The only area for potential improvement involves the need for a little bit of editing to improve flow and clarity of the text. For example, the 6th sentence in the abstract is a little cryptic (These effects were Notch-....). Only after reading the paper does it become clear (given that most people will only read the abstract, it should be tweaked to clarify. The sentence near the bottom of page five (When multiciliated cells ...) could be improved (for example, "Many multiciliated cells in E18.5 *Ahr*^{-/-} mutant tracheae had a lower density or patchy distribution of cilia as compared to analogous cells in wild type embryos").

REVIEWERS' COMMENTS:

Reviewer #1 (Remarks to the Author):

The manuscript by Villa et al has improved through additional analyses and clarification. My only comment is that the field of multiciliated cell transcription is rapidly evolving, with the addition of Gemc1 and Tp73 as important players. It is not necessary that the authors address these new players in the results, but they should be at least referenced and incorporated, as part of the Discussion. In addition, the first reported example of a transition phenotype in the mouse was that described in the Myb mutants, and this needs to be referenced on lines 107.

Our reply: We have now included reference to and citation of the respective papers in the introduction in lines 50-51 and in results in lines 115-117 (previously 107-108).

Reviewer #2 (Remarks to the Author):

This article breaks new ground in understanding the role of the AHR in various cellular processes. The authors have added a considerable amount of new data where possible to more firmly support the conclusions and thus the manuscript is significantly improved. The authors have meaningfully address most of the key comments of the reviewers.

Our reply: Thanks for this positive assessment.

Reviewer #3 (Remarks to the Author):

As this review is for a resubmission, I will forgo summarizing the manuscript. It is, however, important to note that the findings in this manuscript are novel and of general interest. The manuscript has been improved substantially since first submission. For example, some of the new figures are clearer. Also, the phenotype is described in greater detail with respect to aspects associated with cell fate (in the mouse) and to morphogenesis or fidelity of differentiation (in the mouse and in *Xenopus*). The new experiments and the new text really improve the paper. For example, the new text on page 5 helps to clarify the phenotype and the new text on page 7 establishes a role for ARNT. These are nice additions to the story. Indeed, all of the new data has improved the story substantially and the paper is now both exciting and logically presented. The discussion is very well done and puts the work in context. The rebuttal letter is both respectful and comprehensive. Indeed, the experiments in the rebuttal letter should be included in the supplementary data as they really add to the story.

Our reply: We have now added the figures from the rebuttal letter as supplementary figure 2 (single morpholino knock-downs) and Fig. 4f (CCNO reporter constructs in 3T3 cells with or without AhR overexpression) and mention them in the main text (lines 128-130 and 155-157, respectively).

The only area for potential improvement involves the need for a little bit of editing to improve flow and clarity of the text. For example, the 6th sentence in the abstract is a little cryptic (These effects were Notch-...). Only after reading the paper does it become clear (given that most people will only read the abstract, it should be tweaked to clarify. The sentence near the bottom of page five (When multiciliated cells ...) could be improved (for example, "Many multiciliated cells in E18.5 *Ahr*^{-/-} mutant tracheae had a lower density or patchy distribution of cilia as compared to analogous cells in wild type embryos").

Our reply: We have now streamlined the text and modified the indicated phrases in lines 28-31 and 112-114 to simplify and clarify.